# Intermediate soil acidification induces highest nitrous oxide emissions

**Yunpeng Qiu** [1], **Yi Zhang**[1], **Kangcheng Zhang**[1], **Xinyu Xu**[1], **Yunfeng Zhao**[1], **Tongshuo Bai**[1], **Yexin Zhao**[1], **Hao Wang**[1], **Xiongjie Sheng**[1,2], **Sean Bloszies**[3], **Christopher J. Gillespie** [3], **Tangqing He**[1], **Yang Wang**[4], **Huaihai Chen**[5], **Lijin Guo**[6], **He Song**[7], **Chenglong Ye**[1], **Yi Wang**[8], **Alex Woodley**[9], **Jingheng Guo** [10], **Lei Cheng** [11], **Yongfei Bai** [4], **Yongguan Zhu** [12,13,14], **Sara Hallin** [15], **Mary K. Firestone**[16,17] **& Shuijin Hu** [3] ✉

Global potent greenhouse gas nitrous oxide ($N_2O$) emissions from soil are accelerating, with increases in the proportion of reactive nitrogen emitted as $N_2O$, i.e., $N_2O$ emission factor (EF). Yet, the primary controls and underlying mechanisms of EFs remain unresolved. Based on two independent but complementary global syntheses, and three field studies determining effects of acidity on $N_2O$ EFs and soil denitrifying microorganisms, we show that soil pH predominantly controls $N_2O$ EFs and emissions by affecting the denitrifier community composition. Analysis of 5438 paired data points of $N_2O$ emission fluxes revealed a hump-shaped relationship between soil pH and EFs, with the highest EFs occurring in moderately acidic soils that favored $N_2O$-producing over $N_2O$-consuming microorganisms, and induced high $N_2O$ emissions. Our results illustrate that soil pH has a unimodal relationship with soil denitrifiers and EFs, and the net $N_2O$ emission depends on both the $N_2O/(N_2O + N_2)$ ratio and overall denitrification rate. These findings can inform strategies to predict and mitigate soil $N_2O$ emissions under future nitrogen input scenarios.

Nitrous oxide ($N_2O$) is the dominant anthropogenic ozone-depleting substance[1] and is also a long-lived potent greenhouse gas[2]. It has a global warming potential about 265–298 times that of carbon dioxide ($CO_2$) and contributes approximately 7% to the overall global warming[3,4]. Although the $N_2O$ concentration in the atmosphere is low at ca. 330 ppb[5], it is increasing at an accelerating rate of ca. 0.75-1.0 ppb per year[6] because human activities have greatly increased the input of reactive nitrogen (N) in the environment[7,8]. Agricultural N fertilization, in particular, dominates human-induced $N_2O$ emissions[5,8,9]. Since the proportion of reactive N (Nr) emitted as $N_2O$ (i.e., the emission factor, EF) is relatively stable[10,11] in neutral soils, the rate of fertilizer N applied has been considered a robust predictor of $N_2O$ emission. Therefore, the Intergovernmental Panel on Climate Change (IPCC) uses 1% as the default EF of soils at pH of 6.76 (i.e., IPCC default Tier-1) in estimating $N_2O$ emissions[10]. However, both process-based models and atmospheric inversion studies have recently demonstrated that $N_2O$ EFs have significantly increased, which reflects accelerating global $N_2O$ emissions in recent decades[5,6,12]. This suggests that N-application rates are not reliable predictors of $N_2O$ emissions.

Increases in $N_2O$ EFs have been attributed to the non-linear response of soil $N_2O$ emissions to N input[6,12,13], building on the premise that high N input exceeds plant N needs and leads to surplus N for microbial $N_2O$ production[12]. Nitrogen applications further induce a higher proportion of N losses via $N_2O$ in acidic soils[12,14,15] and it is well established that acidity (pH < 5.0) in soil increases the product ratio of $[N_2O/(N_2O + N_2)]$ during denitrification[16–18]. One hypothesized explanation is that pH interferes with the assembly of the $N_2O$ reductase[17]. However, it was also recently shown that soil pH only exerts a control of denitrification product ratio in fertilized soils, while in unfertilized soils, biological controls were more important[15]. Despite increases in the $N_2O/(N_2O + N_2)$ product ratio of denitrification at low pH, $N_2O$ emissions are often low under acidic conditions because acidity

suppresses microbial processes that generate $N_2O$[18–20]. In general, raising soil pH through liming to near-neutral level (pH > 6.5) reduces $N_2O$ emissions, but raising pH in acidic soils (pH < 5.6) to moderately acidic levels (pH = 5.6–6.0) often increases $N_2O$ emissions[18,21–23]. Taken together, these results suggest that soil pH exerts a critical, nonlinear control over $N_2O$ emissions[12,14,24], highlighting the urgency for a comprehensive, mechanistic understanding of pH effects on soil microorganisms and microbial processes that modulate $N_2O$ dynamics.

Soil $N_2O$ emissions originate mainly from two microbial processes, ammonia oxidation being the first step in nitrification, and denitrification, which is the reduction of nitrate to gaseous N (Supplementary Fig. 1). Although ammonia oxidation by ammonia-oxidizing archaea (AOA) and bacteria (AOB) control the rate-limiting step of nitrification in most terrestrial ecosystems[25], denitrification plays a more important role in soil $N_2O$ emissions[26,27]. Since the denitrification process is modular[28] with varying genetic capacities for the different reductive steps in the denitrification pathway among denitrifying microorganisms, the composition of the denitrifying community will control $N_2O$ emissions. Of special concern is the proportion of the denitrifying community harboring the *nosZ* gene coding for the $N_2O$ reductase that converts $N_2O$ to $N_2$ as it is the only known sink for $N_2O$ in the biosphere[29] (Supplementary Fig. 1). There are two phylogenetically distinct clades in the *nosZ* phylogeny: *nosZI* and the recently described *nosZII*[30,31]. Not all denitrifiers carry this gene and therefore terminate denitrification with $N_2O$, but there are also non-denitrifying $N_2O$ reducers which often possess *nosZII*[29]. The ratio of denitrification genes, especially *nirK* and *nirS* encoding the known nitrite reductases involved in denitrification, to the *nosZ* gene abundance is often used as an indication of soil $N_2O$ emissions[15,32,33], but its relationship with soil pH remains largely unexplored.

There is a lack of a unifying, conceptual framework of soil pH impacts on denitrifying microorganisms and $N_2O$ EFs, which critically limits our capacity to predict and mitigate $N_2O$ emissions. Here, we address this knowledge gap with two comprehensive, global meta-analyses of $N_2O$ emission fluxes and EFs in 539 fertilization experiments and of the relationships between soil pH, denitrification gene abundance estimates, and $N_2O$ flux data based on 289 field studies. In addition, three field experiments with acid additions were analyzed to further evaluate the effects of manipulating soil acidity to identify relationships between soil pH and $N_2O$ EFs and disentangle the linkages among soil pH, community composition, and activities of denitrifying microorganisms, and $N_2O$ EFs.

## Results and discussion
### Global synthesis of N input and soil pH effects on $N_2O$ emission factors
We first investigated how soil $N_2O$ EFs related to soil pH and the quantity of N input via fertilization by conducting a meta-analysis based on 539 field fertilization experiments, including 5438 observations of $N_2O$ emission fluxes and 3786 EFs records (Fig. 1a; Supplementary Data 1). Data was collected from experiments distributed among croplands, grasslands, and forests across the globe, published between 1980 and 2019.

The field sites cover soil pH (herein all pH values refer to $pH_{(H_2O)}$) ranging from 2.8 to 9.7, with ca. 58% having a pH of 5.5–7.5 (Fig. 1b; Supplementary Fig. 3). The highest $N_2O$ EFs mainly occurred in weak to moderately acidic soils (pH of 5.6–6.5), with an average EF of 1.2% (Fig. 1b, c). While there was a weak but statistically significant, linear relationship between pH and $N_2O$ EFs, this regression only explained 2.0% of the variation in EFs (Supplementary Fig. 4; see Supplementary

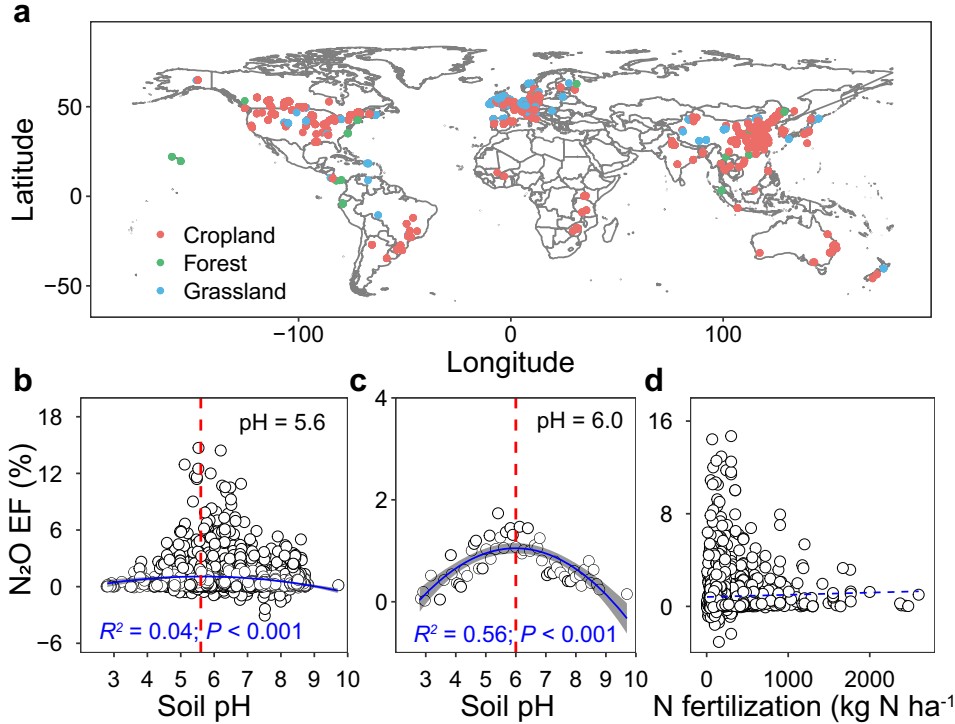

**Fig. 1 | Sample origin and relationship between emission factors (EFs) and soil pH and nitrogen (N) fertilization rates.** Geographic locations of sites included in the meta-analysis (**a**). Relationships between soil pH and coarse EFs (**b**) or averaged EFs (**c**), and the relationship between N fertilization rates and coarse EFs (**d**). A second-order polynomial fit described the hump-shaped relationship between pH and EFs (**b**, EF = −0.0913pH² + 1.030 pH−1.826) or averaged EFs at each pH incremental (**c**, EF = −0.1000 pH² + 1.198 pH−2.537), which reached its maximum at pH = 5.6 (**b**) or 6.0 (**c**), respectively. Linear regression model with two-sided test was used for the statistical analysis (n = 3562 in **b**; n = 58 in **c**; n = 3786 in **d**). The error bands (shaded areas) in (**b** and **c**) represent the 95% confidence intervals around the best-fit regression line, and the blue dashed line in (**d**) indicates an insignificant relationship. Statistics (adjusted $R^2$ and $P$-values) for polynomial regression are indicated. The exact $P$-values: $P < 0.001$ in (**b** and **c**). Source data are provided as a Source Data file.

Table 3 for the model selection). Soil $N_2O$ EFs had a hump-shaped relationship with soil pH, which reached its maximum at pH 5.6 (Fig. 1b; Supplementary Table 3), and explained 4.0% of the variation in $N_2O$ EFs. However, once $N_2O$ EFs were averaged across soil pH in increments (0.1 each), the hump-shaped relationship became markedly more apparent and reached its maximum at pH 6.0 and explained 56% of the variation (Fig. 1c; Supplementary Fig. 5; Supplementary Table 4). These results suggest that interactions between EF and pH diverge around a pH threshold of 5.6–6.0. By contrast, there was no significant linear relationship between $N_2O$ EFs and the quantity of N input (Fig. 1d; Supplementary Table 3). Indeed, the averaged EFs gradually increased with N input and reached their highest around 500–600 kg N ha$^{-1}$ (EF = 1.4%; Fig. 1d; Supplementary Fig. 6). However, the average EFs decreased and remained relatively low in studies with an N input over 600 kg N ha$^{-1}$ (EF = 1.0%; Fig. 1d; Supplementary Fig. 6). These results are inconsistent with the common belief that high N input or soil N content induces high EFs and reconfirm that N quantity alone cannot sufficiently predict $N_2O$ EFs[6,12,24]. Further, the $N_2O$ EFs were significantly higher in acidic tropical soils (pH = 5.5; EFs = 1.1%) than in neutral subtropical (pH = 6.7; EFs = 0.9%) and temperate (pH = 6.9; EFs = 0.8%) soils (Fig. 2a, b), despite significantly lower N input in tropical (170 kg N ha$^{-1}$) than subtropical (223 kg N ha$^{-1}$) and temperate (207 kg N ha$^{-1}$) soils (Fig. 2c). Nevertheless, in tea plantations, all on acidic soils and with high N input (mean = 401 kg N ha$^{-1}$), $N_2O$ EFs positively correlated with both soil pH (Fig. 2d) and the quantity of N input (Fig. 2e), indicating that high acidity reduces $N_2O$ emissions. Additionally, our regression analysis showed that soil organic carbon (SOC) content was negatively correlated with soil pH (Supplementary Fig. 7a; $R^2$ = 0.11; $P < 0.001$), but SOC itself was not significantly related to $N_2O$ EFs (Supplementary Fig. 7b), suggesting that SOC may only indirectly affect $N_2O$ EFs via soil pH. Moreover, although $N_2O$ EFs significantly correlated with mean annual precipitation (MAP), total soil nitrogen (TN), and sand and clay contents, these correlations only explained a low percentage (1–3%) of the

variation in $N_2O$ EFs (Supplementary Fig. 8). Unlike the hump-shaped relationships observed between soil pH and EFs, our further analyses did not find any significant non-linear relations between $N_2O$ EFs and MAP, or sand and clay contents (Supplementary Fig. 8; Supplementary Table 3). There was a hump-shaped relationship between $N_2O$ EFs and TN, but it only explained 2% of the variation of $N_2O$ EFs (Supplementary Fig. 8; Supplementary Table 3). Taken together, these results indicate that although adequate N levels are required for $N_2O$ production, either by nitrification or denitrification, and that multiple soil and climatic factors may affect $N_2O$ emissions, soil pH exerts a dominant, non-linear control over $N_2O$ EFs.

## Soil acidification effects on soil N-cycling microorganisms and $N_2O$

To disentangle the potential microbial mechanisms governing effects of soil pH per se on $N_2O$ EFs, we conducted three field experiments in unfertilized grasslands in which acidity was manipulated (Supplementary Fig. 9). Since none of the experimental sites had received any significant reactive N input (neither N deposition nor N fertilizers)[34–36], the selection pressure of human-derived N on soil N-cycling microorganisms was negligible. We examined how changes in soil pH (i.e., soil acidification) influenced soil available N, abundance of nitrifier and denitrifier functional groups, and soil $N_2O$ emission potential. These experiments were located in three grassland sites with different initial soil pH: a Tibetan alpine meadow (pH = 6.0) near Maqu County, Gansu Province, and a Mongolian steppe (pH = 7.3) in the Xilin River Basin of Inner Mongolia, North China, and a Yellow Loess semi-arid grassland (pH = 8.0) near Guyuan, Ningxia in West China (Supplementary Fig. 9). Each site had a no-acid control (A0) and four levels of acid additions (A1, A2, A3 and A4).

Acid addition consistently reduced soil pH, effectively generating a pH gradient at each site: from 6.0 to 4.7 in the Tibetan alpine soil (Supplementary Fig. 10a), from 7.3 to 4.7 in the Mongolian steppe soil (Supplementary Fig. 10b), and from 8.0 to 7.0 in the Loess soil

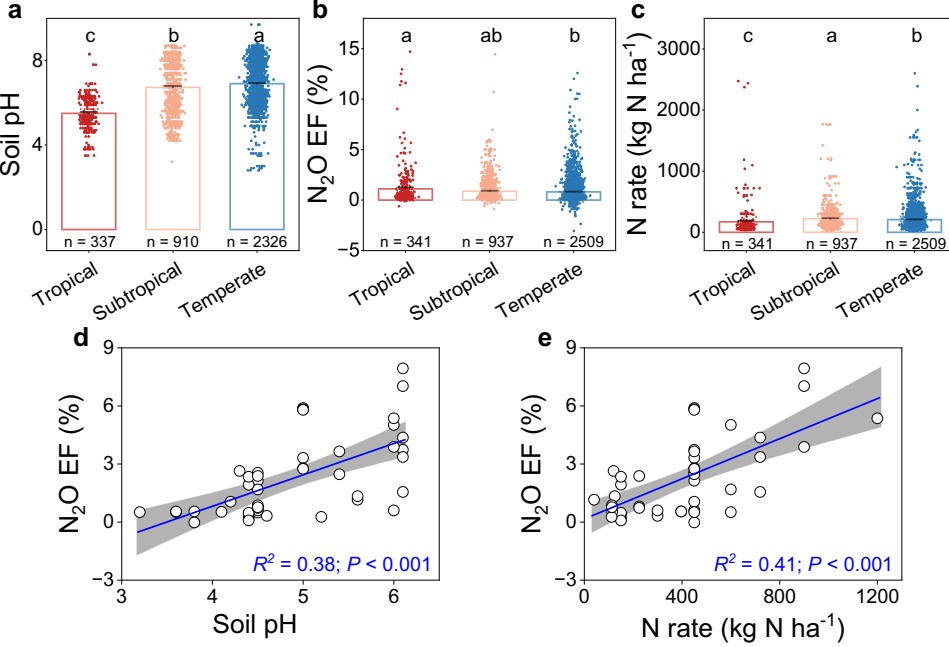

**Fig. 2 | Soil pH, $N_2O$ EF, N fertilization rate in tropical, subtropical and temperate regions, and tea plantations.** Average soil pH (**a**), $N_2O$ emission factors (EFs) (**b**), N fertilization rates (**c**) in different climate zones, and relationship between $N_2O$ EFs and soil pH (**d**) or N fertilization rates (**e**) in tea plantations. In **a**–**c** bars represent mean ± s.e.m and the sample size 'n' represents independent samples. In **a**–**c** different letters indicate a statistical significance of the effect based on non-parametric Wilcoxon test ($P < 0.05$) using the Benjamini and Hochberg (BH) method. The error bands (shaded areas) in (**d** and **e**) represent the 95% confidence intervals around best-fit regression line. Statistics (adjusted $R^2$ and $P$ values) for linear regression are indicated. The exact $P$ values: $P < 0.001$ in (**a**, **c**), and $P = 0.027$ in (**b**). Source data are provided as a Source Data file.

(Supplementary Fig. 10c). Soil $NH_4^+$-N (Supplementary Fig. 11a–c) decreased, but $NO_3^-$-N (Supplementary Fig. 11d–f) increased with increasing soil pH. The abundances of AOA and AOB also increased with increasing soil pH (Supplementary Fig. 12) across the three sites, indicating that soil acidification inhibited AOA and AOB, and nitrification. Similar to AOA and AOB, abundances of *nirK*-, *nirS*- and *nosZI*-type denitrifiers generally increased with soil pH at all three sites, although they were lower in the sandy, low-C Mongolian soil than other two sites (Supplementary Figs. 13 and 14). The *nosZI*-denitrifiers were relatively less sensitive to low soil pH than those with *nirS* or *nirK*, but were more abundant under high soil pH, particularly in the alkaline Loess soil (Supplementary Fig. 13g–i). Soil pH significantly impacted $N_2O$ emissions, which were highest in weakly to moderately acidic soils (pH = 5.6–6.3; Fig. 3e–h). Across the pH gradients at the three sites, we observed hump-shaped relationships between soil pH and the (*nirK* +*nirS*)/*nosZI* ratio, and $N_2O$ emissions, which both peaked at pH = 6.0 (Fig. 3).

We further quantified the potential denitrification activity in the grassland soils under non-limited N- or C-conditions. Incubations with and without addition of acetylene to block the conversion of $N_2O$ to $N_2$ by $N_2O$ reductase allowed us to assess the potential $N_2O$ emission and the direct effect of soil pH on $N_2O$ reduction. Acid additions in the field experiments reduced the denitrification potential in acidic soils but increased it in alkaline soils, leading to the highest denitrification rates in neutral soils (Fig. 4a; Supplementary Fig. 15a–c). As expected, the

$N_2O/(N_2O + N_2)$ product ratio of denitrification decreased as soil pH increased (Fig. 4b; Supplementary Fig. 15d–f)[18,37]. Similar to the relationship between soil pH and the denitrifier community composition, and $N_2O$ emissions (Fig. 3d, h), we observed a hump-shaped relationship between soil pH and potential denitrification (Fig. 4a). However, the pH optimum for potential denitrification (pH = 6.7; Fig. 4a) was higher than that detected for $N_2O$ emissions (pH = 6.0; Fig. 3h). As denitrification rates are often higher under neutral to weak alkaline conditions[20], this difference suggests that decreased pH may have contributed to relatively higher net $N_2O$ emissions by weakening the $N_2O$ sink strength. Collectively, results from the three field experiments provide direct evidence that soil pH modulates the strength of the soil as a $N_2O$ source or sink, mainly because weak to moderate soil acidity promoted $N_2O$ emissions through favoring $N_2O$-producing over $N_2O$-consuming denitrifiers, as well as suppressing reduction of $N_2O$ to $N_2$.

## Global relationship between soil pH and denitrifying microorganisms

To further examine the generality of the relationship between soil pH and the relative composition of the denitrifying microorganisms identified in our acidity manipulation experiments, we conducted a second global meta-analysis to examine the relationship between soil pH and the abundance of denitrification genes in 289 field studies (Fig. 5a). Our dataset covers 3899 gene abundance estimates paired with $N_2O$ flux

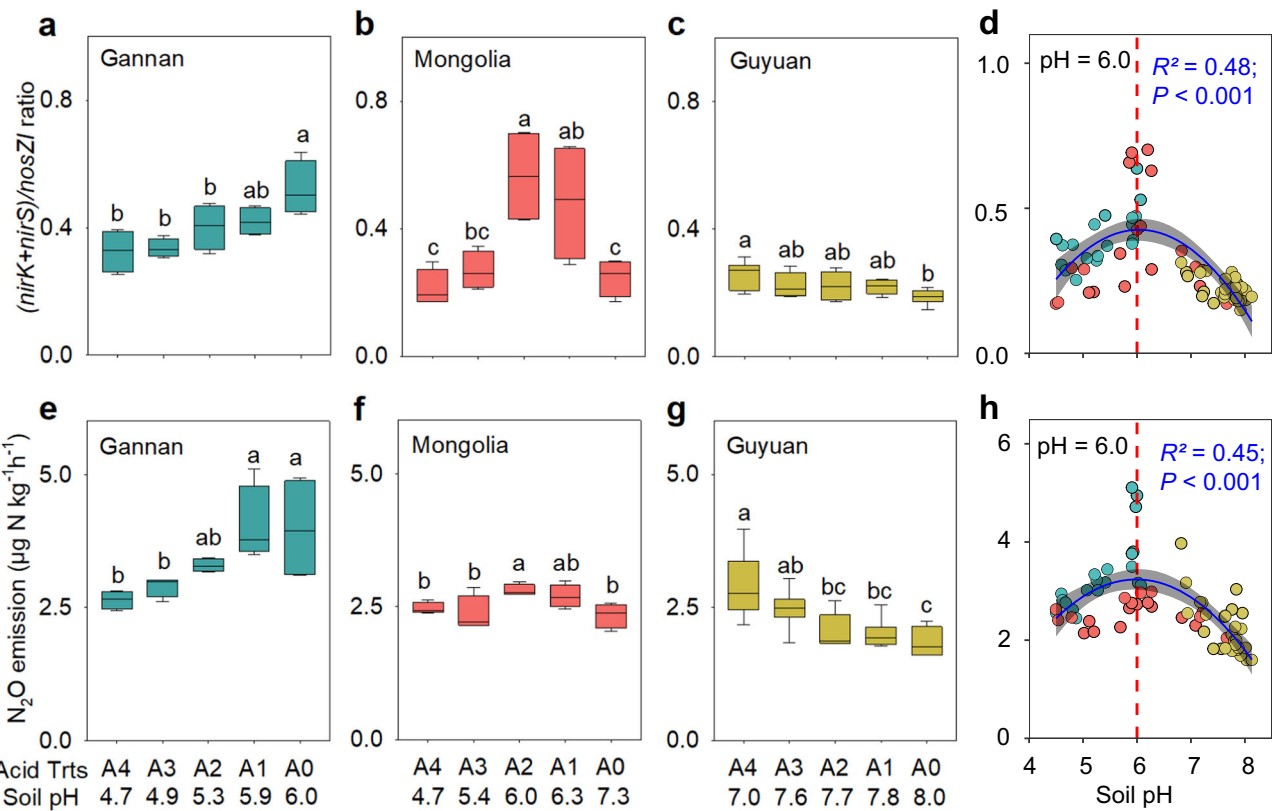

**Fig. 3 | Effects of soil pH on denitrifier community composition and $N_2O$ emissions.** Denitrification gene ratios [(*nirK*+*nirS*)/*nosZI* ratio] (**a–c**) and $N_2O$ emissions (**e–g**) in soil with different pH due to acid treatment from the Gannan alpine meadow, the Inner Mongolia steppe and the Guyuan semi-arid grassland, and the relationship between soil pH and the (*nirK*+*nirS*)/*nosZI* ratio (**d**) and $N_2O$ emissions (**h**) across the three sites. Acid treatments correspond to A0, A1, A2, A3 and A4. A second-order polynomial fit described the hump-shaped relationship between soil pH and the (*nirK*+*nirS*)/*nosZI* ratio (**d**, (*nirK*+*nirS*)/*nosZI* = −0.072pH² + 0.871 pH − 2.204) and $N_2O$ fluxes (**h**, $N_2O$ = −0.357pH² + 4.273 pH − 9.560) across the three sites. In **a–c** and **e–g** one-way ANOVA with two-sided and post-doc test

was conducted to determine significant differences. Different letters indicate a significant difference among acid addition treatment levels at $P < 0.05$. The box plots show the first and third quartiles (box limits), median (center line), and whiskers extend to a maximum of 1.5 times the interquartile range (IQR). For **a**, **b**, **e**, **f**, $n = 4$; **c**, **g**, $n = 6$. The error bands (shaded areas) in (**d** and **h**) represent the 95% confidence intervals around the best-fit regression line. Statistics (adjusted $R^2$ and $P$ values) for polynomial regression are indicated. The exact $P$ values: $P = 0.005$ in **a**, **e**, $P = 0.003$ in **b**, $P = 0.047$ in **c**, $P = 0.019$ in **f**, and $P < 0.001$ in (**g**). Source data are provided as a Source Data file.

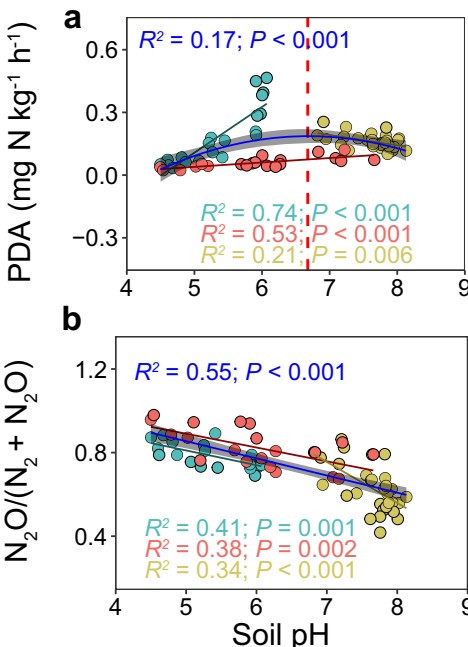

**Fig. 4 | Relationship between soil pH and denitrification.** Correlation between soil pH and potential denitrification activity (PDA) (**a**), and the N$_2$O/(N$_2$O + N$_2$) product ratio of denitrification (**b**) across the Gannan alpine meadow, the Inner Mongolia steppe, and the Guyuan semi-arid grassland. Linear regression model with two-sided test was used for the statistical analysis. A second-order polynomial fit described the hump-shaped relationship between soil pH and PDA across the three field sites (**a**, PDA = −0.0334pH$^2$ + 0.446 pH − 1.304). A linear regression was fitted between soil pH and N$_2$O/(N$_2$O + N$_2$) ratio across the three field sites (**b**, N$_2$O/(N$_2$O + N$_2$) = −0.082 pH + 1.265). Cyan, red, yellow, and blue lines represent correlations in Gannan ($n = 20$), Inner Mongolia ($n = 20$), Guyuan ($n = 30$), and all three sites ($n = 70$), respectively. The error bands (shaded areas) represent the 95% confidence intervals around best-fit regression line. Statistics (adjusted $R^2$ and $P$ values) for polynomial (**a**) and linear (**b**) regression are indicated. Source data are provided as a Source Data file.

data in croplands (796 for *nirK*, 754 for *nirS*, 784 for *nosZI*), grasslands (317 for *nirK*, 330 for *nirS*, 309 for *nosZI*), and forests (234 for *nirK*, 181 for *nirS*, 194 for *nosZI*) (Fig. 5a; see Supplementary Data 3 for detail). Since we only found nine studies with data on *nosZ* clade II combined with N$_2$O emission data from field experiments, only *nosZ* clade I was considered in the following analyses. A positive relationship between soil N$_2$O emissions and the (*nirK*+*nirS*)/*nosZI* ratio across the 289 studies was observed (Supplementary Fig. 16), underscoring the importance of the relationship between microbial sources and sinks for net N$_2$O emissions. The meta-analysis largely supported our manipulation experiments by showing a hump-shaped (unimodal) relationship between soil pH and the abundances of *nirK*- and *nirS*-type denitrifiers, which reached their maximum at pH = 6.0–6.3 (Fig. 5b, c) and pH = 6.3–6.8 (Fig. 5d, e), respectively. However, soil pH was not significantly correlated with either the coarse (Fig. 5f) or averaged (Fig. 5g) abundance of *nosZI*. Consequently, the (*nirK*+*nirS*)/*nosZI* ratio also showed a hump-shaped relationship with soil pH, reaching its maximum at pH of 6.0–6.1 (Fig. 5h, i). These results illustrate that weak to moderately acidic soils generally favor N$_2$O-producing over N$_2$O-consuming denitrifiers and induce high N$_2$O emissions across the global scale.

## A new conceptual framework of soil pH effects on N$_2$O EFs and emissions

Based on the results from the two global meta-analyses and our pH manipulation experiment, we propose that differential effects of soil pH on the denitrification product ratio (i.e., N$_2$O/(N$_2$O + N$_2$)) and

overall denitrification potential jointly control the non-linear responses of EFs to N fertilization (Fig. 6). Thus, the net N$_2$O emission from denitrification depends on both (i) the N$_2$O/(N$_2$O + N$_2$) product ratio of denitrification and (ii) the overall rate of denitrification[18,38], and quantitatively, net N$_2$O emission equals the product of these two parameters. However, both parameters vary distinctly in relation to soil pH (Figs. 4 and 6). In highly acidic soils (pH <5.5), the conversion of N$_2$O to N$_2$ is typically restrained by inhibiting the activity or, as previously hypothesized, the assembly of the N$_2$O reductase[17,18], resulting in high N$_2$O/(N$_2$O + N$_2$) product ratio of denitrification[20,37]. However, low pH often suppresses growth and activity of both nitrifiers and denitrifiers[20,37,39,40], thereby limiting the magnitude of N$_2$O production[37] and leading to low N$_2$O EFs and N$_2$O emission despite a high N$_2$O/(N$_2$O + N$_2$) product ratio of denitrification (Fig. 6). Neutral (pH = 6.6–7.3) and slightly alkaline soils (pH = 7.4–7.8) are optimal for nitrification and denitrification[20,25], but the activity of the N$_2$O reductase is also at its maximum in this pH range, promoting reduction of N$_2$O into N$_2$[18,20]. By contrast, in moderately to weakly acidic soils (pH = 5.6–6.5), both nitrification and denitrification occur at intermediate levels[20,32], and a high (*nirK*+*nirS*)/*nosZI* ratio allows high N$_2$O production but low N$_2$O consumption, leading to high N$_2$O EFs (Fig. 6). Overall, these differential effects of soil pH on N$_2$O-producing and consuming microorganisms, and on N$_2$O reduction result in the highest N$_2$O EFs and emissions in moderately acidic soils.

Our findings that soil pH controls non-linear responses of N$_2$O emissions to N input challenge the prevailing understanding of what regulates N$_2$O EFs. First, soil acidity as the primary determinant of EFs presents a new mechanistic understanding of the recent acceleration of global N$_2$O emissions[14]. Emerging evidence has recently shown that this acceleration was primarily related to high N$_2$O EFs in China and Brazil[5,6], although the underlying mechanisms or causes remained largely unresolved. Our results suggest that high N fertilization rates and its associated soil acidification, especially in China[41], may have jointly contributed to the increased N$_2$O EFs[5]. The high EF in Brazil remains unexplained because average N application rates there are significantly lower than the global average[6,42]. However, one unique, but overlooked, factor is that croplands in Brazil are strongly acidic[43], and liming is frequently applied to raise soil pH to ca. 6.0 for optimal crop growth[44], which might, as our results suggest, have induced high N$_2$O EFs. Second, our findings showing the highest EFs in moderately acidic soils (pH = 5.6–6.0) indicate that the current calculations using the default IPCC EF 1% at pH 6.76 critically underestimate current soil N$_2$O emissions. In general, soil acidification has occurred in a large proportion of agricultural soils in China, US, and Europe because of long-term N fertilization[41,45,46]. However, the degree of acidification varies locally, which can have different effects on soil N$_2$O emissions. According to our results, N fertilization will induce increased acidification and N$_2$O EFs in soils with weak acidity (pH = 6.0–6.7). Moreover, in several Chinese regions, a considerable proportion of agricultural soils are already highly acidic (4.5 < pH < 5.5), where low pH may indeed inhibit N$_2$O emissions (Fig. 6). However, the high acidity is suppressive to the growth of crop plants, and farmers therefore often increase soil pH through liming, which may increase N$_2$O emissions[23]. For neutral or alkaline soils (pH > 6.7), particularly those soils with high buffering capacity, N$_2$O emissions are likely less affected because N fertilization may not significantly reduce soil pH over the short term. This is relevant in light of the expected increase in the world population, especially in tropical and subtropical countries where the major population increase will occur, but current N application rates are low[47,48]. Soils in these regions are typically characterized by low soil fertility and they are moderately to strongly acidic[43]. Increasing plant-available soil N in these regions will therefore be required to ensure crop productivity and economic profits but will inevitably increase N$_2$O EFs and N$_2$O emissions.

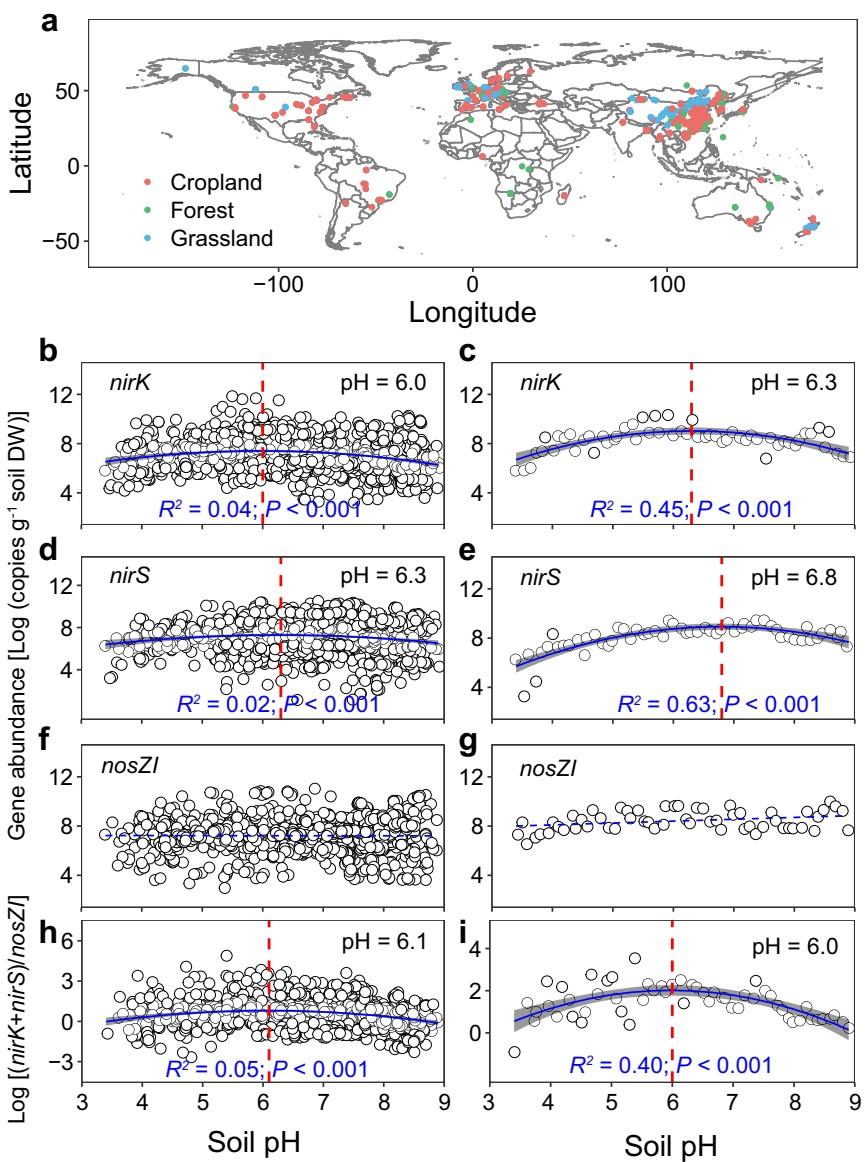

**Fig. 5 | Meta-analysis of pH effects on denitrifying microorganisms.** Geographic locations of the study sites included in the meta-analysis (**a**), and relationships between soil pH and abundances of *nirK*- (**b**, **c**), *nirS*- (**d**, **e**), *nosZI*-type denitrifiers (**f**, **g**), and [(*nirK*+*nirS*)/*nosZI*] ratio (**h**, **i**). Linear regression model with two-sided test was used for the statistical analysis. A second-order polynomial fit described the relationship between soil pH and *nirK* (**b**, **c**), *nirS* abundances (**d**, **e**), and (*nirK*+*nirS*)/*nosZI* ratio (**h**, **i**), which reached the maximum at pH = 6.0 or 6.3 [Log(*nirK*) = −0.129pH$^2$ + 1.546 pH + 2.782 or −0.276pH$^2$ + 3.481 pH − 1.956], pH = 6.3 or 6.8 [Log(*nirS*) = −0.114pH$^2$ + 1.429 pH + 2.833 or −0.281pH$^2$ + 3.816 pH − 4.026], and pH = 6.1 or 6.0 [Log[(*nirK*+*nirS*)/*nosZI*] = −0.115pH$^2$ + 1.398 pH − 3.442 or −0.220pH$^2$ + 2.627 pH − 5.838], respectively. The error bands (shaded areas) in (**b**–**e**, **h**, and **i**) represent the 95% confidence intervals around best-fit regression line, and the blue dashed line in (**f** and **g**) indicates an insignificant relationship. Statistics (adjusted $R^2$ and *P*-values) for polynomial regression are indicated. The exact *P*-values: $P < 0.001$ in (**b**–**e**, **h** and **i**). Source data are provided as a Source Data file.

To conclude, our results indicate that soils with high $N_2O$ EFs (Figs. 1b and 6) significantly overlap in their pH range with pH optima for most crops (pH = 5.5–6.5)[49]. This overlap presents a daunting challenge for $N_2O$ mitigation through manipulating soil pH, highlighting the need for alternative approaches to reduce $N_2O$ emissions. Liming is a common practice in agriculture to reduce toxicity of soil acidity on crop plants[44]. As low soil pH induces high $N_2O$ emission product ratio ($N_2O:N_2$) of denitrification[17,50], raising soil pH to ca. 6.5 has been proposed as a management tool to reduce $N_2O$ emissions[50–52]. However, liming is often economically costly, and farmers tend to only raise soil pH to 5.5–6.0[53,54], which may, based on our results (Figs. 1 and 6), enhance $N_2O$ emissions. Liming also increases soil $CO_2$ emission[23,54], offsetting its impact on $N_2O$ emissions. Our results highlight the urgency to identify alternative approaches that are practically feasible and conducive to lowering $N_2O$ emissions[50] and suggest that manipulation of the community composition and activities of $N_2O$-producing and $N_2O$-consuming microbes may provide a promising approach for $N_2O$ mitigation. Several unique microbial guilds that dominantly control the $N_2O$ sink strength[55] have recently been identified, which may be targeted to reduce the denitrification product ratio[15]. For example, some $N_2O$ reductase-carrying bacteria have adapted to highly acidic soils with pH as low as 3.7[56] and it may be possible to introduce these bacteria into soil to mitigate $N_2O$ emissions in highly acidic soils. However, whether those $N_2O$ reductase-carrying bacteria can be introduced into slightly acidic soils to effectively mitigate $N_2O$ emissions warrants further assessment. In addition, manipulation of $N_2O$-reducing microorganisms might be achieved through crop breeding or cover crop selection because some plants

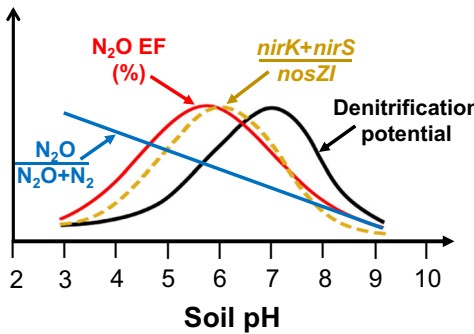

**Fig. 6 | A conceptual framework illustrating the relationships between soil pH and the denitrification product ratio, N₂O emission factor (EF), denitrifier community composition, and the denitrification potential.** The denitrification product ratio [i.e., N₂O/(N₂O + N₂)] is the proportion of denitrification terminating with N₂O, and the N₂O EF is the proportion of fertilizer nitrogen (N) emitted as N₂O (%). The denitrifier community composition is expressed as the ratio between the abundances of N₂O-producing (*nirK*+*nirS*) and N₂O-consuming (*nosZI*) microorganisms. Soil denitrification potential is usually expressed in mg N kg⁻¹ h⁻¹.

produce root exudates and/or plant metabolites inhibiting nitrifying[57,58] and denitrifying[59] microorganisms. Further, reducing access of nitrifiers to ammonium through manipulating N sources (e.g., slow-releasing fertilizers)[60], supporting nitrate ammonifiers reducing nitrate to ammonium[61,62], and enhancing plant N uptake, and/or inhibiting nitrifiers (e.g., nitrification inhibitors) can decrease N₂O emissions from both nitrification and denitrification[63]. Overall, our study provides compelling evidence illustrating that there is a hump-shape relationship between soil pH and N₂O EF, leading to highest N₂O emissions under moderate soil acidity. These findings suggest that raising pH through liming has limited capacity for N₂O mitigation due to multiple biological and economic constraints, and that direct manipulation of N₂O-producing and N₂O-consuming microbes may provide novel approaches for N₂O mitigation under future reactive N input scenarios.

## Methods

### Meta-analysis 1 of global synthesis of N input and soil pH effects on N₂O emission factors (N₂O EFs)

The data collection and analysis followed the preferred reporting items for systematic reviews and meta-analyses (PRISMA) guidelines (see Supplementary Fig. 2a for further information). We conducted an extensive search for studies of N fertilization and soil N₂O emissions published between 1980 and 2019 through the Web of Science, Google Scholar, and the China Knowledge Resource Integrated Database (http://www.cnki.net/). The keywords were used: (i) "nitrogen addition" OR "nitrogen deposition" OR "nitrogen amendment" OR "nitrogen fertilization"; (ii) "soil" OR "terrestrial"; and (iii) "N₂O" OR "nitrous oxide". We also extracted data and re-evaluated all studies from the databases published by Stehfest and Bouwman[11], Liu and Greaver[64], Shcherbak et al.[12], Liu et al.[65], Wang et al.[14], Charles et al.[66], Deng et al.[67], Maaz et al.[68], Cui et al.[24], and Hergoualc'h et al.[69]. In order to avoid selection bias, we extracted peer-review publications with the following criteria: (a) only field studies in which the control and N fertilization treatment sites were located under the same climate, vegetation and soil conditions were included; (b) only chamber-based field experiments conducted in croplands, forests and grasslands were included; (c) studies using nitrification inhibitors were excluded. This yielded a dataset of 5438 observations of N₂O emission fluxes from 539 field studies that spanned 42 countries and 570 sites (Fig. 1a; please see Supplementary Data 1). Experiments were grouped into three regions based on absolute latitude: tropical (23.4 °S–23.4 °N), subtropical (23.4–35.0 °S or °N), and temperate (>35.0 °S or °N). For each study, soil properties (i.e., pH, clay,

silt and sand content, organic carbon, and total nitrogen) and climate (i.e., mean annual precipitation (MAP) and temperature (MAT)) were directly obtained either from texts and/or tables or extracted from figures using the GetData Graph Digitizer software (ver. 2.22, http://www.getdata-graph-digitizer.com).

Nitrogen fertilization rates and soil N₂O emissions obtained from the literature were converted into the unit of kg N ha⁻¹, respectively. Fertilizer-induced N₂O emission was then calculated as the difference in soil N₂O emission between the fertilization treatment (E_N) and the no-fertilizing control (E_O). Then the emission factor (EF) of N₂O emissions of each fertilization treatment was calculated as the percentage of N₂O emission relative to the amount of N fertilization rate (see Eq. 1). This yielded a dataset of 3786 N₂O EF values (please see Supplementary Data 1).

$$EF(\%) = 100 \times \frac{E_N - E_O}{N} \qquad (1)$$

To determine the impact of soil pH on N₂O EF, pH was divided into 58 groups by 0.1 unit (pH: 2.8–9.7). Soil pH was measured in water in most studies, but it was measured in CaCl₂ or KCl in solution in a small number of experiments. We converted soil pH values measured in CaCl₂ or KCl into water-based soil pH values, following the method described by Henderson and Bui[70] and Kabala et al.[71], respectively. A few studies did not specifically state the reagent used, and we assumed that water was used there. Notably, soil acidity or alkalinity was divided into: ultra-acidic of pH < 3.5, extremely acidic of pH = 3.5–4.4, very strongly acidic of pH = 4.5–5.0, strongly acidic of pH = 5.1–5.5, moderately acidic of pH = 5.6–6.0, slightly acidic of pH = 6.1–6.5, neutral of pH = 6.6–7.3, slightly alkaline of pH = 7.4–7.8, moderately alkaline of pH = 7.9–8.4, and strongly alkaline of pH = 8.5–9.0, following the Soil Science Division Staff (2017)[72].

One major issue in the method using the coarse EFs is that the pH increments with more data points are given higher weight than the pH increments with fewer data points. Consequently, the statistical analysis is highly skewed towards the pH increments with a large number of field experiments and measurements. However, this does not provide a fair assessment of the pH effect on N₂O EFs. Therefore, we adopted the average method by averaging all the N₂O EFs at each pH increment to obtain the mean EF and then giving all the pH increments equal weights. We followed the method used by Linquist et al.[73] and Feng et al.[74] to evaluate the mean EF for the different pH groups (Eqs. (2) and (3)).

$$M = \frac{\sum (Y_i \times W_i)}{\sum (W_i)} \qquad (2)$$

$$W_i = \frac{n}{o} \qquad (3)$$

We used Eq. (2) to calculate the weighted mean values for each pH unit group. In Eq. (2), M is the mean value of EF. $Y_i$ is the observation of EF at the ith pH unit group. $W_i$ is the weight for the observations from the ith pH unit group and was calculated with Eq. (3), in which n is the replicates in each field experiment for each study, and o is the total number of observations from the ith pH unit group. At a given pH increment, this approach of weighting assigned more weight to well-replicated field measurements, reporting more precise EF (%) estimates[73,74].

### Field experiments of soil pH manipulations and their effects on denitrifiers and their activities

Reactive N input affects N-cycling microbes and N₂O emissions directly by increasing N availability for nitrification and denitrification and indirectly by inducing soil acidification. In order to determine the

direct impact of soil acidification, we manipulated soil pH through adding diluted acids to create a pH gradient in three grassland experiments in the Tibetan Plateau, Inner Mongolian Plateau, and the Yellow Loess Plateau in China. We choose grasslands for three reasons. First, we wanted to assess the effect of soil pH without confounding effects of N fertilization. Unlike most Chinese croplands that have received high amounts of N fertilization[41], these grasslands are located in remote areas where there was low ambient N deposition and no N fertilization assuring minimal impact of human-derived N on soil N-cycling microbes[34–36]. Second, since none of the experimental grasslands had received any significant reactive N input (N deposition or N fertilizers), the selection pressure of human-derived N on soil N-cycling microorganisms was negligible. Third, we wanted to have field experiments on acidic, neutral, and alkaline soils that also have decent amounts of available soil N. Available soil N (particularly $NO_3^-$) in other unfertilized soils, like forest soils, is very low and likely constrains N-cycling microbes[75]. Moreover, grasslands potentially contribute 20% of total $N_2O$ flux to the atmosphere at the global scale[76,77]. A considerable proportion of global grasslands are under moderate to intensive management, and it is expected that more grasslands will be under fertilization, likely increasing $N_2O$ emissions[77].

The three acid addition experiments were established in three grasslands with distinct climatic and soil conditions (Supplementary Table 1; see Supplementary Data 2). The first experiment was set up in an alpine meadow at Gansu Gannan Grassland Ecosystem National Observation and Research Station (33°59′N, 102°00′E, ca. 3538 m a.s.l.) in Maqu county, Gannan Prefecture, Gansu Province, China. Over the last forty years, the MAP and MAT at this site were at 620 mm and 1.2 °C, respectively. The soil was categorized as Cambisol (FAO taxonomy) and moderately acidic with a pH value of ca. 6.0 with moderate pH buffering capacity[35]. The second experiment took advantage of an existing study on a steppe ecosystem at the Inner Mongolia Grassland Ecosystem Research Station of the Chinese Academy of Sciences (43°38′N, 116°42′E, 1250 m a.s.l.) near Xilin city, Inner Mongolia, China. The MAT at this site was 0.3 °C with the lowest in January (−21.6 °C) and the highest in July (19.0 °C). It has had a MAP of 346.1 mm with the majority (ca. 80%) occurring in summer (June to August). It had a dark chestnut soil (Calcic Chernozem according to ISSS Working Group RB, 1998) with a nearly neutral pH value (ca. 7.3) and with high sand content and low pH buffering capcity[34]. The third experiment was in a semi-arid grassland at the Yunwu Mountains Natural Preserve (36°10′−36°17′N, 106°21′−106°27′E, 1800–2100 m a.s.l.) on the Loess Plateau, Guyuan, Ningxia, Northwest China. This site has a typical semiarid climate, and the mean annual rainfall was about 425 mm with about two-thirds (60–75%) falling in July-September. Over the last three decades, this site had a MAT of 7.0 °C (the lowest in January at −14 °C and the highest in July at 22.8 °C). The soil was a montane gray-cinnamon type classified as a Calci-Orthic Aridisol or a Haplic Calcisol in the Chinese and FAO classification, and alkaline with a pH of 8.0 and high pH buffering capacity[36].

At each site, a single factor of acid (sulfuric acid) addition experiment was designed. To minimize any potential direct acid damage to living plants and soil organisms, the specific dose of concentrated sulfuric acid (98%) needed for each plot was first diluted into 60 L of tap water and then sprayed into each plot. Equal amounts of water only were added to the no-acid controls (A0).

At the Gannan alpine site, the acid addition experiment was established in 2016 with five levels of acid addition[78]: 0 (the control, A0), 1.32 (A1), 5.29 (A2), 9.25 (A3), and 14.53 (A4) mol H⁺ m⁻² yr⁻¹. Twenty plots (2 m × 2 m each) were then arranged in a randomized block design including four replicate blocks separated by 1 m buffer zones. Diluted sulfuric acid solution was applied twice each year (half of the designed dosage each time) in early June and late September of 2016, late April and late September of 2017, and late April 2018. At the Inner Mongolia steppe site, the acid experiment was initiated in 2009

with seven gradients of acid addition[79]: 0, 2.76, 5.52, 8.28, 11.04, 13.80, and 16.56 mol H⁺ m⁻² yr⁻¹. The experiment was randomly positioned in a block design with 5 replicate blocks, leading to a total of 35 field plots (2 m × 2 m each). Diluted acid solution at the designed concentration was added to each plot in early September 2009, early June 2010, and early September 2010. Soil pH in all treatments stabilized and no additional acid has been added since 2010[79]. For this study, we randomly chose four replicate field plots of five treatments, 0 (the control, A0), 2.76 (A1), 5.52 (A2), 11.04 (A3), and 16.56 (A4) mol H⁺ m⁻² yr⁻¹, to investigate the impact of soil acidification on soil nitrifiers, denitrifiers and denitrification. The acid experiment at the Guyuan site was established in 2016 with 30 plots (2 m × 2 m each) using a randomized block design[80]. It had five levels of acid additions with six replicate blocks separated by 1 m walkways. The five levels of acid additions were: 0 (the control, A0), 0.44 (A1), 1.10 (A2), 7.04 (A3), and 17.61 (A4) mol H⁺ m⁻² yr⁻¹, respectively. Diluted acid solution was applied twice each year (half each time) in early June and late September of 2016, late April and late September of 2017, and early May 2018.

In mid-August 2018 when plant biomass peaked, three soil cores (5.0 cm dia.) were collected at 0–10 cm depth from each plot at both Gannan and Guyuan sites, and then mixed to form a composite sample per plot. For the Inner Mongolia site, soil samples were collected in the same way in early September 2020. Composited soil samples collected in field were placed on ice in coolers, and sent by express mail to the laboratory in Nanjing, China. All soil samples were first sieved through a mesh (2 mm) to remove rocks and dead plant materials. A small subsample (ca. 50 g) of each field soil sample was immediately stored at −20 °C for molecular analyses, and the remainder was kept at 4 °C in the refrigerator for later chemical and microbial analyses that were all initiated within 2 weeks. Soil pH in a soil-to-water (1:5, w/w) slurry was measured on an Ultramete-2 pH meter (Myron L. Company, Carlsbad, CA, USA). Inorganic $NH_4^+$-N and $NO_3^-$-N were extracted with 0.5 M $K_2SO_4$, and their concentrations in the extracts were quantified on a continuous flow injection auto-analyzer (Skalar SAN Plus, Skalar Inc., The Netherlands)[35]. For each soil sample, 0.3 g (dry soil equivalent) frozen soil was used to extract total genomic DNA with PowerSoil DNA kits (MoBio Laboratories, Carlsbad, CA, USA). The DNA quantity and quality were determined by a Nanodrop spectrophotometer (Thermo Scientific, Wilmington, DE, USA). The copy numbers of AOA-*amoA*, AOB-*amoA*, *nirK*, *nirS*, and *nosZI* genes were determined using the Real-Time quantitative PCR System (Applied Biosystems, Foster City, CA, USA). The primer sets of *crenamoA*23F/*crenamoA*616r (ATGGTCTGGCTWAGACG/GCCATCCATCTGTATGTCCA)[81], *amoA*-1F/*amoA*-2R (GGGGTTTCTACTGGTGGT/CCCCTCGGAAAGCCTTCTTC)[82], *nirK*876/*nirK*1040 (ATYGGCGGVAYGGCGA/GCCTCGATCAGRTTRTGG TT)[83], *nirS*Cd3aF/*nirS*R3cd (AACGYSAAGGARACSGG/GASTTCGGRT GSGTCTTSAYGAA)[84], and *norZ*1f/*norZ*1R (WCSYTGTTCMTCGAGC-CAG/ATGTCGATCARCTGVKCRTTYTC)[85] were used for the amplification of AOA-*amoA*, AOB-*amoA*, *nirK*, *nirS*, and *nosZI* gene, respectively. Each qPCR reaction (20 μL volume) was performed with 10 μL SYBRs Premix Ex Taq™ (Takara, Dalian, China), 1 μL template DNA corresponding to 8–12 ng, 0.5 μL of each primer, 0.5 μL bovine serum albumin (BSA, 5 mg mL⁻¹) and 7.5 μL distilled deionized $H_2O$ (dd$H_2O$). The standard curve for determining the gene copy number was developed using the standard plasmids of different dilutions as a temperate. The standard plasmids were generated from the positive clones of the 5 target genes, which were derived from the amplification of the soil sample[55]. The amplification efficiency of the qPCR assays ranged from 90 to 100% with $R^2 > 0.99$ for the standard curves. We checked potential qPCR reaction inhibition via the amplification of a known amount of the pGEM-T plasmid (Promega) with T7 and SP6 primers, adding to the extracts of DNA samples or water. No amplification reaction inhibitions in the samples were detected.

We did not directly monitor soil $N_2O$ fluxes in the field, mainly because the field sites were remote. Instead, microcosm incubation

experiments were conducted to determine potential soil N$_2$O emissions. For each soil sample, field soil (20.0 g dry mass equivalent) was placed into a 125-mL dark bottle, and deionized water was added to adjust soil moisture to ca. 70% water-filled pore space (WFPS), creating a moisture condition conducive for denitrifiers and denitrification[33,86]. The high soil moisture content favored anaerobic processes since O$_2$ diffusion into the soil was restricted and effects of oxygen should be negligible. All bottles were loosely covered with fitting lids and incubated in a dark incubator at 20 °C. It is worth mentioning that both nitrification and denitrification processes produce N$_2$O, but optimum N$_2$O emissions from denitrification often occur at 70–80% WFPS[33,86]. Also, our results showed that soil pH had a linear relationship with soil nitrifiers (Supplementary Fig. 12) and the high soil moisture suppressed nitrification. Thus, the design of the incubation experiments targeted N$_2$O from anaerobic processes like denitrification, and N$_2$O emissions from nitrification or other aerobic processes were not considered[33].

To determine the N$_2$O emissions, gas samples were taken from the headspaces of the incubation bottles as described by Zhang et al. [35]. More specifically, all incubation bottles were flushed with fresh air (2 min each) prior to the gas sampling, then immediately sealed and incubated for 6 h in the dark. A gas sample of 15 mL was taken from the headspace of each incubation bottle and was immediately transferred into a vial for gas chromatograph (GC) measurement. After gas sampling, all incubation bottles were loosely covered until the next gas sampling to ensure minimum water loss. Gas sampling was conducted 5 times, respectively, at 12, 24, 48, 72, and 96 h after the incubation initiation. N$_2$O concentrations in the sampling vials were determined within 24 h after the sampling collection on a GC equipped with an electron capture detector (ECD) (GC-7890B, Agilent, Santa Clara, CA, USA). The N$_2$O fluxes were calculated using the formula[35]:

$$F = \rho \times V \times \Delta C \times \frac{273}{(273 + T) \times W} \qquad (4)$$

where F is the soil N$_2$O gas flux rates (μg N kg$^{-1}$ soil h$^{-1}$), $\rho$ is the standard state gas density (kg m$^{-3}$), V is the bottle volume (L), $\Delta C$ is the difference in N$_2$O concentration (ppm) between two samples (0 and 6 h), T is the incubation temperature at 20 °C, and W is the dry weight of soil (kg).

We further determined soil potential denitrification activities (PDA), using the modified acetylene (C$_2$H$_2$) inhibition technique[55,87]. For each field soil sample, two sub-samples (each 5.0 g dry soil equivalent) were respectively put into two 100 mL sterile serum bottles. Then, 8 mL of N- and C-containing solution (KNO$_3$ at 50 mg NO$_3^-$-N g$^{-1}$ dry soil, glucose, and glutamic acid, each at 0.5 mg C g$^{-1}$ dry soil) was added to create a soil slurry conducive for denitrification. To measure the PDA, 10% C$_2$H$_2$ was injected into one bottle to inhibit N$_2$O reductase activity so that the N$_2$O produced was not reduced to N$_2$. In the other bottle, no C$_2$H$_2$ was added so that all enzymes of denitrification remained active and the N$_2$O detected was the net difference between the production and consumption of N$_2$O[55]. All serum bottles were incubated in dark at 25 °C with agitation at 180 rpm. Gas samples (10 mL) were taken from the headspace at 2, 4 and 6 h after the beginning of the incubation for determination of N$_2$O concentrations on a GC (GC-7890B, Agilent, Santa Clara, CA, USA).

## Meta-analysis 2 of relationships between soil pH and N$_2$O-producing or N$_2$O-consuming denitrifying microorganisms

Similar to Meta-analysis 1, the data collection and analysis were also carried out according to the PRISMA guidelines (Supplementary Fig. 2b). We conducted an extensive search in Web of Science and Google Scholar for studies in which nirK-, nirS- and nosZ (clade I and II) had been quantified with the two sets of search terms: (1) nirK,

nirS or nosZ gene, and (2) soil or terrestrial. In total, the search resulted in ca. 1539 article hits in December 2021. All articles were carefully read through to select those based on field studies, whereas those based on microcosm studies were excluded. There were 286 published papers that met our criteria. We also included the data from the three field acid addition experiments described above. Special attention was also directed towards checking whether nosZ clade I, nosZ clade II or both were quantified. Only 26 published studies quantified nosZII and, among these, only nine also reported soil N$_2$O emissions in the field (see Supplementary Data 3 for detail). Therefore, the gene nosZ in the dataset in this study only refers to nosZ clade I. Thus, the final dataset contained data from 501 sites reported by 289 studies, and included 1347, 1265, and 1287 abundance estimates of nirK, nirS and nosZI genes, respectively (see Supplementary Data 3 for detail).

We extracted data either from tables, texts or from figures using the GetData Graph Digitizer software (ver. 2.22; http://getdata-graph-digitizer.com). For each article, we extracted the following information for our analysis: the abundance of nirK, nirS, nosZI and nosZII genes (copy numbers per g soil), soil pH, and depth of collected soils. Latitude, altitude, MAP and MAT of the experimental sites were also recorded. All information of N$_2$O emissions (N$_2$O emission rates and/or cumulative N$_2$O emissions) was extracted. Because various publications reported the results of N$_2$O emissions in different units, we converted all N$_2$O emission rates into the unit of μg N m$^{-2}$ h$^{-1}$. Data were log-transformed to meet statistical tests assumptions (if necessary). In the literature, most data of gene abundances were presented in the form of log-transformed numbers, we first transformed them back to real numbers and obtained the average gene abundances for each pH increment, and then again log-transformed. Similar to Meta-analysis 1, we examined the relationships between soil pH and abundances of denitrifying microorganisms, using both coarse abundance and averaged abundance of each functional group of denitrifiers at each pH increment.

## Statistical analyses

In Meta-analysis 1, we examined potential linear or quadratic relationships between N$_2$O EFs and soil pH, MAP, MAT, soil sand, silt and clay content, SOC or TN. In Meta-analysis 2, we examined potential linear and quadratic relationships between soil pH and the abundance of nirK-, nirS-, nosZI-type denitrifiers, or the (nirK+nirS)/nosZI ratio. The model goodness of fit was evaluated with the Akaike information criterion (AICc) where a lower AICc value represents a model with a better fit[88,89]. In general, differences in AICc higher than 2 indicate that models are substantially different[88]. Information on the AICc index was obtained using the package MuMIn from R[90]. Given the large number of samples included in the meta-analyses, we interpreted the statistical significance of individual predictors using a conservative α of 0.001 following model selection by AICc. In Meta-analysis 1, we did a non-parametric alternative of Kruskal–Wallis analysis together with Pairwise Wilcox test to determine the differences in soil pH, N$_2$O EFs, and N fertilization rate among different climate zones.

For the field experiments, we used linear mixed-effects (LME) models to determine the effects of acid addition on the response variables at each site, treating the acid treatments as fixed effects and block as a random effect. One-way analysis of variance (ANOVA) followed by Duncan's multiple-range tests were used to compare the means among acid addition levels across all response variables. Then, we examined the relationships between soil pH and N$_2$O emissions, PDA, N$_2$O/(N$_2$O + N$_2$) ratio or the (nirK+nirS)/nosZI ratio across all three field sites, using linear or quadratic regression. We used the Akaike information criterion (AICc) to evaluate the model's goodness of fit[88]. All analyses were conducted in R (version 4.1.1)[91].

## Reporting summary

Further information on research design is available in the Nature Portfolio Reporting Summary linked to this article.

## Data availability

The data used in this study are available in Supplementary Data 1–3.xlsx and online in the Figshare database (https://doi.org/10.6084/m9.figshare.24591522).

## Code availability

The code is available in the Figshare database (https://doi.org/10.6084/m9.figshare.24591522).

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

## Acknowledgements

We are thankful to the staff at the three experimental sites for maintaining the long-term field experiments. We also thank Dima Chen, Guozhen Du, Hui Guo, Jiuxin Guo, Zhen Li, Xi Luo, Fanglong Su, Fuwei Wang, Peng Wang, Yanan Wei, Xuebin Yan, Qilai Yang, Hao Zhang, Juanjuan Zhang, Qingzhou Zhao and Xianhui Zhou for assistance with field acid addition treatments. This research was partially supported by Natural Science Foundation of China (NSFC) [Grant Nos. 32001140 (Y.Q.), 32371626 (Y.Q.) and 32171553 (Yi Z.)], China Postdoctoral Science Foundation [Grant No. 2022T150325 (Y.Q.)], and USDA-National Institute of Food and Agriculture (NIFA) [Grant No. 2018-51106–28773 (S.H.)].

## Author contributions

S.H., Y.Q., and Yi.Z. conceived the research and designed the study. Y.Q. conducted the meta-analyses. Yi Z., Y.Q., and S.H. conceived the conceptual diagram. Y.Q., X.X., K.Z., X.S., T.B., Yi W., Yi Z., Y.W., and Y.B. maintained the three field experiments. Y.Q., X.X., Yi.Z., K.Z., Yf.Z., Ye. Z., H.W., and T.H. conducted the experiments and performed the lab analyses. Y.Q., K.Z., Yi Z., S.H., and H.C. performed the data analyses. Y.Q. and S.H. wrote the first draft of the manuscript. S.B., C.J.G., L.G., H.S., C.Y., A.W., J.G., L.C., Y.Z., S.H., and M.K.F. reviewed and edited the draft.

## Competing interests

The authors declare no competing interests.

## Additional information

¹College of Resources and Environmental Sciences, Nanjing Agricultural University, Nanjing 210095, China. ²School of Ecology and Environmental Sciences, Yunnan University, Kunming 650091, China. ³Department of Entomology & Plant Pathology, North Carolina State University, Raleigh, NC 27695, USA. ⁴State Key Laboratory of Vegetation and Environmental Change, Institute of Botany, Chinese Academy of Sciences, Beijing 100093, China. ⁵State Key Laboratory of Biocontrol, School of Ecology, Shenzhen Campus of Sun Yat-sen University, Shenzhen, Guangdong 518107, China. ⁶International Magnesium Institute, College of Resources and Environment, Fujian Agriculture and Forestry University, Fuzhou 350002, China. ⁷College of Agronomy, Anhui Agricultural University, Hefei 230036, China. ⁸State key Laboratory of Loess and Quaternary Geology, Institute of Earth Environment, Chinese Academy of Sciences, Xi'an 710061, China. ⁹Department of Crop and Soil Sciences, North Carolina State University, Raleigh, NC 27695, USA. ¹⁰Beijing Key Laboratory of Farmland Soil Pollution Prevention and Remediation, College of Resources and Environmental Sciences, China Agricultural University, Beijing 100193, China. ¹¹MOE Key Laboratory of Biosystems Homeostasis & Protection, College of Life Sciences, Zhejiang University, Hangzhou 310058, China. ¹²Key Laboratory of Urban Environment and Health, Institute of Urban Environment, Chinese Academy of Sciences, Xiamen 361021, China. ¹³State Key Laboratory of Environmental Chemistry and Ecotoxicology, Research Center for Eco-Environmental Sciences, Chinese Academy of Sciences, Beijing 100049, China. ¹⁴Zhejiang Key Laboratory of Urban Environmental Processes and Pollution Control, CAS Haixi Industrial Technology Innovation Center in Beilun, Ningbo 315830, China. ¹⁵Department of Forest Mycology and Plant Pathology, Swedish University of Agricultural Sciences, Uppsala 75007, Sweden. ¹⁶Department of Environmental Science, Policy, and Management, University of California, Berkeley, Berkeley, CA 94720, USA. ¹⁷Earth and Environmental Sciences, Lawrence Berkeley National Laboratory, Berkeley, CA 94720, USA. ✉e-mail: shuijin_hu@ncsu.edu

