## [Peer Review File · Nature Communications]

Intermediate soil acidification induces highest nitrous oxide emissionsREVIEWER COMMENTS

Reviewer #1 (Remarks to the Author):

Based on the meta-analysis and results from the pH manipulation experiment, the authors in this manuscript found that the pH as the master factor controlled the N₂O EF, and very interestingly, they found that a hump shaped relationship between pH and N₂O index and also (nirS+nirK)/nosZI. Most important thing is that the authors challenged the well accepted knowledge that increasing the pH by liming is the main method for mitigate the N₂O emission. Instead, they proposed a strategy for mitigating the N₂O emission from acidic agriculture soil by regulating the N₂O producing and consuming microbial guilds. The findings in this manuscript is important to comprehensively understand the pH effect on the N₂O emission in the farm land soil.

Here are some specific comments:

Since that the nosZ II carrying bacteria often act as the super N₂O sink and their N₂O mitigation effect are better than the nosZ I carrying bacteria, Only discussed the (nirK+nirS)/nosZI ratio in the manuscript would somehow mislead.

Since that the pH optima for most crops are ranged from 5.5 to 6.5, and this pH range also is the maximum N₂O EF being found. We may conclude that the N₂O emission in the low pH acidic soil could be attributed to the inhibition of N₂O reductase by the low pH condition. Is it possible to introduce the low pH adapted N₂O reductase carrying bacteria, which were reported in previous study (Lycus et al., ISME J. 2017 Oct; 11(10): 2219–2232.) to the low or moderately low pH agricultural soil for the purpose of N₂O mitigating ?

In the microcosm incubation experiment, During the incubation between the gas sampling, the vials were not sealed to allow the gas exchange. However, the incubation with oxygen may inhibit the growth and activity of N₂O sink bacteria. Will this influence the results for N₂O EF respond to the varied pH? It is a little strange that the hump-shaped relationship appeared for PDA and pH. What is the reason of decrease of PDA along with the increase of pH in Guyuan semi-arid grassland soil? Should be the improper measurement of N₂O?

Reviewer #2 (Remarks to the Author):

The authors conducted two extensive global meta-analyses studying the effect of soil pH on N₂O EFs (with >5400 observations) and on denitrifier community (with >3800 observations), along with field pH manipulation experiments on three grassland sites. The findings suggested that soil pH primarily governs N₂O EFs and emissions by affecting the denitrifier community composition, with the highest N₂O EF observed in soils with moderately acidic pH. Their efforts are to be commended. The findings are crucial in managing optimal pH levels in agricultural soils to ensure nutrient availability and overall soil health. However, several aspects need further clarification and consideration.

The potential effects of other factors that could drive N₂O emissions/EFs should be considered. For example, it is well known that soil moisture affects nitrification, denitrification, and their contribution to N₂O emissions, and that soil moisture (also rainfall) is quite variable and could lead to sporadic N₂O emissions. I wonder how these effects were factored into the relationship between soil pH and N₂O EFs? Moreover, the pH buffering capacity of soils, dependent on various soil properties, should not be overlooked, as it could affect the rate of soil acidification in response to acid additions.

The inclusion of the pH manipulation experiments on three grasslands is valuable but inconclusive. It is unclear why only grassland sites were selected, especially when the two global meta-analyses included cropping and forest systems as well. The representation of grasslands as compared to other ecosystems concerning their response to acid addition (soil acidification) and associated microbial response and N₂O emissions warrants justification. In particular, the authors stated that none of the grassland sites had received significant reactive N input (neither N deposition nor N fertilizers) (lines 162-163). In this case, the expected N₂O emissions from fertilizer N would be minimal. This scenario differs greatly from cropping systems, which often receive much higher N input.

The authors projected a global increase in N₂O emissions, attributing it to rising N₂O EFs due to fertilizer-induced acidification, mainly in China, Europe, and the US (lines 253-255). However, this generalization may oversimplify the situation. I would expect a diverse range of soil pH in these countries. Based on the hump-shaped relationship illustrated in this study, for acidic soils in these regions, further acidification would lead to lower N₂O EFs (not necessarily higher as predicted by the authors).

Please elaborate why the patterns of coarse EFs and averaged EFs against soil pH were so much different (Fig. 1). Are there any underlying mechanisms that should be further explored?

Point by point responses to Reviewers' Comments on Manuscript
Nature Communications (NCOMMS-23-36237A)
(Comments in black and Responses in blue)

REVIEWER COMMENTS

Reviewer #1 (Remarks to the Author):

Based on the meta-analysis and results from the pH manipulation experiment, the authors in this manuscript found that the pH as the master factor controlled the N₂O EF, and very interestingly, they found that a hump shaped relationship between pH and N₂O index and also (nirS+nirK)/nosZI. Most important thing is that the authors challenged the well accepted knowledge that increasing the pH by liming is the main method for mitigate the N₂O emission. Instead, they proposed a strategy for mitigating the N₂O emission from acidic agriculture soil by regulating the N₂O producing and consuming microbial guilds. The findings in this manuscript is important to comprehensively understand the pH effect on the N₂O emission in the farm land soil.

Reply: Many thanks for your supports.

Here are some specific comments:

Since that the nosZ II carrying bacteria often act as the super N₂O sink and their N₂O mitigation effect are better than the nosZ I carrying bacteria, Only discussed the (nirK+nirS)/nosZI ratio in the manuscript would somehow mislead.

Reply: Very insightful comments. Yes, we are aware that *nosZII* carrying bacteria can be an important N₂O sink (Jones et al. 2014; Hallin et al. 2018). In the revision, we attempted to expand our second meta-analysis by including studies that measured *nosZII* abundance. Unfortunately, however, most studies have so far only measured the abundance of *nosZI*; only 26 field studies reported *nosZII* abundance and only 9 of them reported both *nosZII* abundance and N₂O emissions in field (please see Supplementary Data 2 for more detail). Because of the limited availability of data, it is not appropriate to make any generalization about the linkages among soil pH, *nosZII* and N₂O emissions. Therefore, the only option is to focus on the (*nirK+nirS*)/*nosZI* ratio, which contains over 1,671 paired data points from 289 studies.

Since that the pH optima for most crops are ranged from 5.5 to 6.5, and this pH range also is the maximum N₂O EF being found. We may conclude that the N₂O emission in the low pH acidic soil could be attributed to the inhibition of N₂O reductase by the low pH condition. Is it possible to introduce the low pH adapted N₂O reductase carrying bacteria, which were reported in previous study (Lycus et al., ISME J. 2017 Oct; 11(10): 2219–2232.) to the low or moderately low pH agricultural soil for the purpose of N₂O mitigating?

Reply: Great question and suggestion. Yes, we believe that it is possible. In this revision, we have now discussed how the N₂O reductase-carrying bacteria adapted to low pH might be used in the acidic soil to mitigate N₂O emission (see Line 291-293).

“For example, some N₂O reductase carrying bacteria have adapted to highly acidic soils with pH as low as 3.7 (Lycus et al. 2017) and it may be possible to introduce these bacteria into soil to mitigate N₂O emissions in highly acidic soils.”

In the microcosm incubation experiment, During the incubation between the gas sampling, the vials were not sealed to allow the gas exchange. However, the incubation with oxygen may inhibit the growth and activity of N₂O sink bacteria. Will this influence the results for N₂O EF respond to the varied pH?

Reply: Very good point. Yes, incubation with oxygen may inhibit the growth and activity of N₂O sink bacteria. However, during our microcosm incubation, soil moisture was adjusted to ca. 70% water-filled pore space (WFPS) by adding deionized water, creating a moisture condition conducive for denitrifiers and denitrification (Butterbach-Bahl et al. 2013; Qiu et al. 2019). This high soil moisture content favored anaerobic processes since O₂ diffusion into the soil was restricted and effects of oxygen should be negligible. We have now added more details to the Method section (see Line 471-474).

It is a little strange that the hump-shaped relationship appeared for PDA and pH. What is the reason of decrease of PDA along with the increase of pH in Guyuan semi-arid grassland soil? Should be the improper measurement of N₂O?

Reply: Yes, the hump-shape curve between pH and PDA did not explain a very high variation. This is likely because denitrification is an electron-consuming and heterotrophic process, and available C (here soil organic C, SOC) is another major driving factor of denitrification (Firestone & Davidson 1989). Denitrification is often favored under neutral to weak alkaline conditions (Šimek & Cooper 2002). Although the Mongolia soil has a near-neutral pH, it had significantly lower SOC than two other sites. Therefore, it did not have the highest PDA. In contrast, the Alpine acidic soil of pH 6.0 had a much higher SOC (9.8%) (see Supplementary Table 1), likely contributing to the highest PDA.

In addition, Guyuan soil was an alkaline soil with pH around 8.0, and a small decrease in soil pH slightly increased PDA. Again, this increase was likely because denitrifiers were favored under neutral and weak alkaline conditions.

Reviewer #2 (Remarks to the Author):

The authors conducted two extensive global meta-analyses studying the effect of soil pH on N₂O EFs (with >5400 observations) and on denitrifier community (with >3800 observations), along with field pH manipulation experiments on three grassland sites. The findings suggested that soil pH primarily governs N₂O EFs and emissions by affecting the denitrifier community composition, with the highest N₂O EF observed in soils with moderately acidic pH. Their efforts are to be commended. The findings are crucial in managing optimal pH levels in agricultural soils to ensure nutrient availability and overall soil health. However, several aspects need further clarification and consideration.

Reply: Thank you very much for your positive comments and encouragement. We have now modified our manuscript following your advice and comments.

The potential effects of other factors that could drive N₂O emissions/EFs should be considered. For example, it is well known that soil moisture affects nitrification, denitrification, and their contribution to N₂O emissions, and that soil moisture (also rainfall) is quite variable and could lead to sporadic N₂O emissions. I wonder how these effects were factored into the relationship between soil pH and N₂O EFs? Moreover, the pH buffering capacity of soils, dependent on various soil properties, should not be overlooked, as it could affect the rate of soil acidification in response to acid additions.

Reply: Excellent comments. We have now examined the relationships between N₂O EFs and multiple other climate or soil factors (Supplementary Fig. 7), in addition to soil pH and SOC (Supplementary Fig. 6) described in the previous version. Results from these new analyses are included in Results (see Line 150-155) and also in the Supplementary information (Supplementary Fig. 7). Results showed that although N₂O EFs statistically significantly correlated with mean annual precipitation (MAP), total soil nitrogen (TN), and sand and clay contents, the correlations only explained a very low percentage of the variation in N₂O EFs (1-3%) (Supplementary Fig. 7).

As to the pH buffering capacity, soils with high pH buffering capacity should be more resistant to soil acidification. We have now considered the pH buffering capacity in the Discussion as well as mentioned this in the Method section (see Line 268-271; Line 393; Line 400; Line 407).

Supplementary Fig. 7. Relationship between mean annual precipitation (MAP, a), temperature (MAT, b), soil sand (c), silt (d) and clay content (e), and total soil nitrogen content (TN, f), and N_2O EFs in the meta-analysis of fertilizer N effects on N_2O EFs. The shaded area presents the 95% confidence intervals around the linear regression line. Statistics (R^2 and P values) for linear regression are indicated.

The inclusion of the pH manipulation experiments on three grasslands is valuable but inconclusive. It is unclear why only grassland sites were selected, especially when the two global meta-analyses included cropping and forest systems as well. The representation of grasslands as compared to other ecosystems concerning their response to acid addition (soil acidification) and associated microbial response and N_2O emissions warrants justification. In particular, the authors stated that none of the grassland sites had received significant reactive N input (neither N deposition nor N fertilizers) (lines 162-163). In this case, the expected N_2O emissions from fertilizer N would be minimal. This scenario differs greatly from cropping systems, which often receive much higher N input.

Reply: These are insightful comments and suggestions. For the field experiments, we chose the three grasslands as model systems for three major reasons (see Line 159-163, Line 374-386 and Supplementary Notes):

1) The field manipulative experiments allowed us to examine how acidification (soil pH) alone affects N_2O EFs and N_2O emissions. Input of N fertilizers (often in the form

NH_4^+) directly affects N_2O EFs and emissions by increasing N substrate, but also indirectly affects them through soil acidification, leading to the confounding effects of N and pH (Wang et al. 2018; Jones et al. 2022). Most croplands in China have received high amounts of N fertilization (Guo et al. 2010; Yu et al. 2022) and soil pH in the major Chinese crop-production areas, on average, has declined significantly by ca. 0.5 unit from the 1980s to the 2000s (Guo et al. 2010). Also, plant harvest removes other nutrients from agricultural soils and agricultural practices (e.g., tillage and fungicides) disturb soils, which could have some effects on N_2O EFs and emissions. Thus, isolation of acidification from other human-induced activities on and N_2O EFs is a major reason why we did not use agricultural soils.

2) Applications of N fertilizers modify the community composition of N-cycling microbes (both nitrifiers and denitrifiers) (Ouyang et al. 2018) and the selective pressure of N fertilizer inputs on the functional characteristics of soil microbial communities may mask the effects of soil pH on N-cycling microbes and N_2O emissions (Jones et al. 2022). Since none of the experimental grasslands had received any significant reactive N input (N deposition or N fertilizers), the selection pressure of human-derived N on soil N-cycling microorganisms was negligible.

3) We wanted to have field experiments on acidic, neutral and alkaline soils that also have decent amounts of available soil N. Available soil N (particularly NO_3^-) in other unfertilized soils, like forest soils, is very low and likely constrains N-cycling microbes (Stark & Hart 1997).

Moreover, grasslands cover about 40% of the global land surface and are widely used as pasture, supporting the livelihoods of more than 1.3 billion people through livestock production (O'Mara 2012; Buisson et al. 2022). Global grasslands potentially contribute 20% of total N_2O flux to the atmosphere (Dangal et al. 2019; Chang et al. 2021). Globally, there is a considerable proportion of grasslands under moderate to intensive management with fertilization and it is expected that more grasslands will be under fertilization in the future (Chang et al. 2021).

The authors projected a global increase in N_2O emissions, attributing it to rising N_2O EFs due to fertilizer-induced acidification, mainly in China, Europe, and the US (lines 253-255). However, this generalization may oversimplify the situation. I would expect a diverse range of soil pH in these countries. Based on the hump-shaped relationship illustrated in this study, for acidic soils in these regions, further acidification would lead to lower N_2O EFs (not necessarily higher as predicted by the authors).

Reply: Great comment. We agree that our previous statement “*We therefore forecast continuous increases in global N_2O emissions in the coming decades due to increasing N_2O EFs since N fertilization in temperate regions (mainly in China, Europe and US) will further induce acidification*” were too simplistic. Because of variation in soil pH within each country or region, pH in a specific soil, rather than the average pH of the region, will primarily affect N_2O emissions. Therefore, we have now reorganized the Discussion section based on the relative range of soil pH (see Line 261-271).

“In general, soil acidification has occurred in a large proportion of agricultural soils in China, US and Europe because of long-term N fertilization (Bowman et al. 2008; Guo et al. 2010; Chen et al. 2023). However, the degree of acidification varies locally, which can have different effects on soil N_2O emissions. According to our results, N fertilization will induce increased acidification and N_2O EFs in soils with weak acidity

(pH = 6.0-6.7). Moreover, in several Chinese regions, a considerable proportion of agricultural soils are already highly acidic ($4.5 < \text{pH} < 5.5$), where low pH may indeed inhibit N_2O emissions (Fig. 6). However, the high acidity is suppressive to the growth of crop plants, and farmers therefore increase soil pH through liming, which may increase N_2O emissions (Wang et al. 2021). For neutral or alkaline soils ($\text{pH} > 6.7$), particularly those soils with high pH buffering capacity, N_2O emissions are likely less affected because N fertilization may not significantly reduce soil pH over the short term.”

Please elaborate why the patterns of coarse EFs and averaged EFs against soil pH were so much different (Fig. 1). Are there any underlying mechanisms that should be further explored?

Reply: This difference primarily stems from the fact that these two methods give different statistical weight for each soil pH point. In the coarse EFs, the pH increment with more data points (e.g., pH6.0, 153 data points) is given much higher weight than the pH increment with fewer data points (e.g., pH4.5, 14 data points). Consequently, the statistical analysis is highly skewed towards the pH increments with large number of field experiments and measurements, leading to a weak linear regression that can explain a very low proportion of the data variation (Supplementary Fig. 4; also see Wang et al. 2018). However, this does not provide a fair assessment of the pH effect on N_2O EFs. The averaged method solves this issue by averaging N_2O EF data points at each pH increment to obtain one EF value and then giving all the pH increments equal weights. Our work is the first to address this issue and identifies the unique relationship between soil pH and N_2O EFs that has previously been overlooked.

This issue might have been overlooked in a couple of previous meta-analyses that only showed a weak linear relationship (Wang et al. 2018) or no relationship (Cui et al. 2021). One major reason may be due to the scale-issue in the diagram. Because some N_2O EFs were much higher than others, the Y axis in the coarse-pH diagram must be set much higher than the average to cover all the data points, which could visually give a false impression of linear relationship (Figure R1a, b). If we reduce the Y-axis scale in the pH-coarse diagram to the same range for the pH-averaged diagram, we can visually see that the pattern of the pH-coarse curve and the pH-averaged curve were indeed similar (see Figure R1c, d below). However, the explanatory power ($R^2=0.04$ vs. $R^2=0.57$) differed greatly because the coarse method gives too much weight to the pH points with more measurements, as described in the previous paragraph.

Response Figure R1. Relationship between soil pH and coarse EFs or averaged EFs with the same y axis range [a, b (from -7 to 20) or c, d (from -1 to 4)].

To further validate and demonstrate our elaboration/argument, we created a model dataset using different numbers of data points at each pH value. Then, we calculated coarse and averaged EFs. From the diagram below, we can see that the curve patterns between pH and coarse EFs or averaged EFs were similar (Figure R2), but the two curves are very different in their how much the variance of the dataset is explained ($R^2=0.04$ for the coarse method but $R^2=0.75$ for the average method). That is exactly what happened with our real dataset in our meta-analysis.

Response Figure R2. Relationship between soil pH and coarse EFs (a) or averaged EFs (b) with a model dataset.

References:

1. Buisson, E., Archibald, S., Fidelis, A. & Suding, K. N. Ancient grasslands guide ambitious goals in grassland restoration. *Science* **377**, 594–598 (2022).
2. Bowman, W. D., Cleveland, C. C., Halada, L., Hreško, J. & Baron, J. S. Negative impact of nitrogen deposition on soil buffering capacity. *Nat. Geosci.* **1**, 767–770 (2008).
3. Butterbach-Bahl, K., Baggs, E. M., Dannenmann, M., Kiese, R. & Zechmeister-Boltenstern, S. Nitrous oxide emissions from soils: how well do we understand the processes and their controls? *Philos. Trans. R. Soc. London, Ser. B* **368**, 20130122 (2013).
4. Chang, J. et al. Climate warming from managed grasslands cancels the cooling effect of carbon sinks in sparsely grazed and natural grasslands. *Nat. Commun.* **12**, 118 (2021).
5. Chen, C., Xiao, W. & Chen, H. Y. H. Mapping global soil acidification under N deposition. *Glob. Change Biol.* **29**, 4652–4661 (2023).
6. Cui, X. et al. Global mapping of crop-specific emission factors highlights hotspots of nitrous oxide mitigation. *Nat. Food* **2**, 886–893 (2021).
7. Dangal, S. R. S. et al. Global nitrous oxide emissions from pasturelands and rangelands: magnitude, spatiotemporal patterns and attribution. *Glob. Biogeochem. Cycles* **33**, 200–222 (2019).
8. Firestone, M. K. & Davidson, E. A. in *Exchange of trace gases between Terrestrial Ecosystems and the Atmosphere* (eds Andreae, M. O. & Schimel, D. S.) (Wiley, Chichester, UK, 1989).
9. Guo, J. H. et al. Significant acidification in major Chinese croplands. *Science* **327**, 1008–1010 (2010).
10. Hallin, S., Philippot, L., Löffler, F. E., Sanford, R. A. & Jones, C. M. Genomics and ecology of novel N₂O-reducing microorganisms. *Trends Microbiol.* **26**, 43–55 (2018).
11. Jones, C. et al. Recently identified microbial guild mediates soil N₂O sink capacity. *Nat. Clim. Change* **4**, 801–805 (2014).
12. Jones, C. M., Putz, M., Tiemann M. & Hallin S. Reactive nitrogen restructures and weakens microbial controls of soil N₂O emissions. *Commun. Biol.* **5**, 273. (2022).
13. Lycus, P., Bøthun, K. L., Bergaust, L., Shapleigh, J. P, Bakken, L. R & Frostegård, Å. Phenotypic and genotypic richness of denitrifiers revealed by a novel isolation strategy. *ISME J.* **11**, 2219–2232 (2017).
14. O'Mara, F. P. The role of grasslands in food security and climate change. *Ann. Bot.* **110**, 1263–1270 (2012).
15. Ouyang, Y., Evans, S. E., Friesen, M. L. & Tiemann, L. K. Effect of nitrogen fertilization on the abundance of nitrogen cycling genes in agricultural soils: A meta-analysis of field studies. *Soil Biol. Biochem.* **127**, 71–78 (2018).

16. Qiu, Y. et al. Shifts in the composition and activities of denitrifiers dominate CO₂ stimulation of N₂O emissions. *Environ. Sci. Technol.* **53**, 11204–11213 (2019).
17. Šimek, M. & Cooper, J. E. The influence of soil pH on denitrification: Progress towards the understanding of this interaction over the last 50 years. *Eur J Soil Sci.* **53**, 345–354 (2002).
18. Stark, J. M. & Hart, S. C. High rates of nitrification and nitrate turnover in undisturbed coniferous forests. *Nature* **385**, 61–64 (1997).
19. Wang, Y. et al. Potential benefits of liming to acid soils on climate change mitigation and food security. *Glob. Change Biol.* **27**, 2807–2821 (2021).
20. Wang, Y. Guo, J. Vogt, R. D., Mulder, J., Wang, J. & Zhang, X. Soil pH as the chief modifier for regional nitrous oxide emissions: new evidence and implications for global estimates and mitigation. *Glob. Change Biol.* **24**, e617–e626 (2018).
21. Yu, Z.; Liu, J.; Kattel, G. Historical nitrogen fertilizer use in China from 1952 to 2018. *Earth Syst. Sci. Data* **14**, 5179–5194 (2022).

REVIEWERS' COMMENTS

Reviewer #1 (Remarks to the Author):

All my concern in last review has been properly responded. However, some minor revision is needed for this manuscript before the publication.

The gene copy number of N₂O source and sink is indeed related with the activities of production and consumption of N₂O in the soil. but the quantity of these genes was not the sole determining factor. Even with the same amount of nosZ containing bacteria, the reduction of N₂O is largely lower under low pH (for instance, pH<6) than that in neutral pH due to the malfunction of NOR complex under low pH condition. I totally agree with the concept mentioned in this manuscript, "the net N₂O emission from denitrification depends on both (i) the N₂O/(N₂O+N₂) product ratio of denitrification and (ii) the overall rate of denitrification". This should be recognized as the core idea running through the entire article. It would be better if this also be demonstrated in some way in the abstract of this manuscript.

some specific comments:

The difference between figure 5f and 5g was neither described in the figure legend or in the text.

Reviewer #2 (Remarks to the Author):

I appreciate the authors' effort in conducting additional analyses, which provide further information to substantiate their findings. I have a couple of follow-up questions/suggestions:

1. In the new analysis of the correlations between N₂O EFs and climate/soil factors, the authors mentioned that these correlations explained only a very low percentage of the variation in N₂O EFs (1-3%) (lines 150-152) to support the assertion that soil pH is a key factor. I seek further clarification since, as indicated in lines 128-130, soil pH explains only 4% of the variation in N₂O EFs, which is comparable to the 1-3% range. In fact, the relationship between soil pH and N₂O EFs became evident only when averaged across soil pH in increments, where soil pH explained 57% of the variation (lines 131-132). Slight rephrasing of the newly added sentences might prevent potential confusion or enhance clarity.
2. The authors proposed introducing N₂O reductase-carrying bacteria adapted to highly acidic soils as a mitigation strategy for N₂O emissions in such environments. However, earlier lines suggested that the issue primarily occurs under soil pH of 5.6-6.0 (slightly acidic). Considering this, would such a strategy still be effective for slightly acidic soils, given that bacteria adapted to highly acidic conditions may be stressed by a sudden change (increase) in pH when introduced?

Point by point responses to Reviewers' Comments on Manuscript
Nature Communications (NCOMMS-23-36237B)
(Comments in black and Responses in blue)

REVIEWER COMMENTS

Reviewer #1 (Remarks to the Author):

All my concern in last review has been properly responded. However, some minor revision is needed for this manuscript before the publication.

Reply: Many thanks for your supports. We have now revised our manuscript following your suggestion.

The gene copy number of N₂O source and sink is indeed related with the activities of production and consumption of N₂O in the soil. but the quantity of these genes was not the sole determining factor. Even with the same amount of nosZ containing bacteria, the reduction of N₂O is largely lower under low pH (for instance, pH<6) than that in neutral pH due to the malfunction of NOR complex under low pH condition. I totally agree with the concept mentioned in this manuscript, "the net N₂O emission from denitrification depends on both (i) the N₂O/(N₂O+N₂) product ratio of denitrification and (ii) the overall rate of denitrification". This should be recognized as the core idea running through the entire article. It would be better if this also be demonstrated in some way in the abstract of this manuscript.

Reply: Insightful comment. We have now added the core point that the net N₂O emission depends on both the N₂O/(N₂O+N₂) ratio and overall denitrification rate in the abstract.

some specific comments:

The difference between figure 5f and 5g was neither described in the figure legend or in the text.

Reply: Thanks for pointing this out. We have now described Figure 5f and 5g both in the main text and the Figure 5 legend (see Lines 221-222; Lines 863-875).

Reviewer #2 (Remarks to the Author):

I appreciate the authors' effort in conducting additional analyses, which provide further information to substantiate their findings. I have a couple of follow-up questions/suggestions:

Reply: Thank you very much for your positive comments and encouragement. We have now modified our manuscript following your suggestions.

1. In the new analysis of the correlations between N₂O EFs and climate/soil factors, the authors mentioned that these correlations explained only a very low percentage of the variation in N₂O EFs (1-3%) (lines 150-152) to support the assertion that soil pH is a key factor. I seek further clarification since, as indicated in lines 128-130, soil pH explains only 4% of the variation in N₂O EFs, which is comparable to the 1-3% range. In fact, the relationship between soil pH and N₂O EFs became evident only when averaged across soil pH in increments, where soil pH explained 57% of the variation (lines 131-132). Slight rephrasing of the newly added sentences might prevent potential confusion or enhance clarity.

Reply: Good point. We have now rephrased the sentences (see Lines 149-155) to explain how the relationship between N₂O EFs and soil pH was different from those between N₂O EFs and other soil and climate parameters.

2. The authors proposed introducing N₂O reductase-carrying bacteria adapted to highly acidic soils as a mitigation strategy for N₂O emissions in such environments. However, earlier lines suggested that the issue primarily occurs under soil pH of 5.6-6.0 (slightly acidic). Considering this, would such a strategy still be effective for slightly acidic soils, given that bacteria adapted to highly acidic conditions may be stressed by a sudden change (increase) in pH when introduced?

Reply: Good question. However, whether those N₂O reductase carrying bacteria can be introduced into slightly acidic soils to effectively mitigate N₂O emissions warrants further assessment. We have now revised the sentence (see Line 293-295).